# Observed Recent Change in Climate and Potential for Decay of Norwegian Wood Structures

**Terje Grøntoft**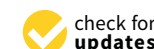

NILU-Norwegian Institute for Air Research, 2027 Kjeller, Norway; teg@nilu.no; Tel.: +47-63-898-023

**Abstract:** The wood rot decay of structures and buildings in Norway represents high costs. This paper reports the observed trends for the potential rot decay of Norwegian wood structures in the cities of Oslo and Bergen over the recent 55 years, calculated as the "wood rot climate index" developed by Scheffer, and compares the reports with previous reported values based on climate change modelling. The observed change in the wood rot climate index was close to the modelling result. Bergen is exposed directly to the westerly Atlantic winds and has among the highest rain amounts in Norway, whereas Oslo is shielded by the Scandinavian mountain chain and has much less rain. The change in the wood rot climate index since 1961 was about 20% in both cities, but the trend in the index (climate index change per year) was about 80% stronger in Bergen. The absolute index changes were largest in the summer; then spring (50 to 60% of the summer increase); and small, zero, or even negative (autumn in Oslo) in the remaining seasons. The relative changes were higher in the spring than summer and very high in Bergen in the winter from a low value. The change to positive index values in the spring and winter indicates temperature and humidity conditions favoring the growth of wood rot and, thus, extended the rot duration through the year. The expected increase in the future wood rot decay potential in Norway shows the need for increased focus on adaption measures to reduce the related damages and costs.

**Keywords:** climate change; climate impact; climate index; wood rot; wood decay; wooden buildings

---

## 1. Introduction

The purpose of this work was to compare the evidence available from meteorological measurements performed over the recent past, with reported modelling predictions for changes in the wood rot decay risk in Norway. The paper reports the calculated values and trends for the climate wood rot index developed by Scheffer [1] for the two most densely built urban areas in Norway, Oslo and Bergen. The calculations used the observed values for temperature and precipitation for a 55 years period, from 1961 to September 2016. The observed trends were compared with predictions for the index based on climate modelling reported by Reference [2]. The importance of changes in this index for the maintenance of wooden buildings is discussed.

The present observed and future expected climate change will impact the performance [3] and increase the decay rate, and thus the maintenance and conservation costs, of built structures and cultural heritage (e.g., [4–9]). Wood is a prominent building material in Norway (and other countries) historically and today. A large fraction of new, in particular residential, buildings and historical buildings, classified as cultural heritage, are made from wood. The decay of the wooden buildings represents both economic loss and a loss of original heritage value. Reference [4] discussed how moisture damage counts for a very high percentage of the total number of building defects in Norway and that the fraction of the total Norwegian building stock which is located in high decay risk areas will increase manifold in this century due to climate change. It is, therefore, "essential that proper adaptation measures are implemented, including [the] use of historical climate data and climate

predictions in planning, design, and construction". The reduction of decay, damage, and maintenance costs by the implementation of proper adaptation measures is likely to be huge. It is, therefore, important to monitor and analyze the changing environment to assess the change in the load on built structures and to verify the climate change predictions.

Reference [10] gives an extensive description and discussion of the historical and future expected Norwegian climate. Some of their main conclusions are that the average annual temperature over mainland Norway has increased with 0.06 to 0.11 °C per decennium since 1900 and with 0.36 to 0.55 °C from the 1971–2000 to the 1985–2014 period, with some variation between regions. The annual precipitation amount has increased with 1.0 to 2.0% per decennium since 1900, with an outlier of 0.2% for the Northern most region of Varanger (at a lower level of significance) and with an average of 4% from the 1971–2000 to the 1985–2014 period but with up to 10% in some regions and up to 15% in some seasons. During this century, it is further expected that the annual average temperature over mainland Norway will increase with 3.3 to 6.4 °C and that the annual precipitation amount will increase with 7 to 23% depending on region, with larger changes in the north than in the south. Other important changes are that episodes with heavy rain will increase; rain floods will become larger and more frequent, whereas snowmelt floods will become smaller and fewer; there will be less snow in the lowlands but more snow in some mountain areas; there will be fewer and smaller glaciers; and the see level will rise with 15 to 55 cm depending on the location. These changes will affect the conditions of the built structures in different ways.

The wood rot grows when the climatic conditions are beneficial for the fungi. The growth mainly happens when the moisture content in the wood is over 20% [11] and the temperature is over 3 °C [1]. Relatively few fungi continue to metabolize cut and worked wood products. These fungi have different metabolizing preferences, and the observed decay is different. The brown rot lives mostly on the cellulose and produces a cracked appearance. The white rot also consumes the lignin, does not crack the wood, and is often spongy. The brown rot is sometimes called "dry rot", as it can itself conduct water into wood constructions. Soft rot is more related to mold and mainly attacks the wood surface (e.g., References [12,13]).

The wood rot decay potential in wood structures due to the ambient climate was calculated by Reference [1] by combining the monthly temperature and precipitation data and summarizing it to an annual climate index ("climate wood rot index"). The development of the index was based on the analysis of "typical growth–temperature curves for a few commonly occurring decay fungi" in the USA. The index results for the decay potential were compared with the experimental rates of decay for the most decay-susceptible tested wood species (western hemlock, white fir, ponderosa pine sapwood, and southern pine sapwood), showing "remarkably similar results". Reference [1] used this climate index to calculate the decay potential of wooden structures in the USA. The index was recently used to calculate the decay potential of wooden structures in Norway in the recent past (1961–1990), and to predict the change to the near (2021–2050) and far (2100) future based on input data from climate change modelling [2,4].

Bergen and Oslo represent very different climates. Bergen is exposed directly to the westerly Atlantic winds and has among the highest rain amounts in Norway. Oslo is shielded from the westerly winds by the Scandinavian mountain chain and has much less rain. The annual precipitation normal for the period 1961 to 1990 was for Bergen 2000–2500 (2250) mm and for Oslo 700–1000 (763) mm precipitation [14].

## 2. Materials and Methods

Temperature and precipitation data from one meteorological station in Oslo and two stations in Bergen were used in the calculations. The monthly mean values for temperature for the Oslo-Blindern (no. 18700) and Bergen-Florida (no. 50540) meteorological stations for all months from January 1961 through to September 2016 were obtained from the Norwegian Meteorological Institute open data source web pages [15]. The data for precipitation measured daily every 12 h at 0600 and 1800 universal

coordinated time (UTC) from January 1961 through to September 2016 at the Oslo-Blindern station, from January 1961 through to December 1982 at the Bergen-Fredriksberg (no. 50560) station, and from January 1983 through to September 2016 at the Bergen-Florida station were, similarly, obtained from Reference [15]. Data from the two stations in Bergen, both located in the center of the city, were used, as the precipitation data were not available for the whole period from one station. The daily precipitation amount for every day (24 h) was obtained by adding the precipitation amount reported at 0600 and 1800 h. Figure 1 shows the locations of the stations in Oslo and Bergen.

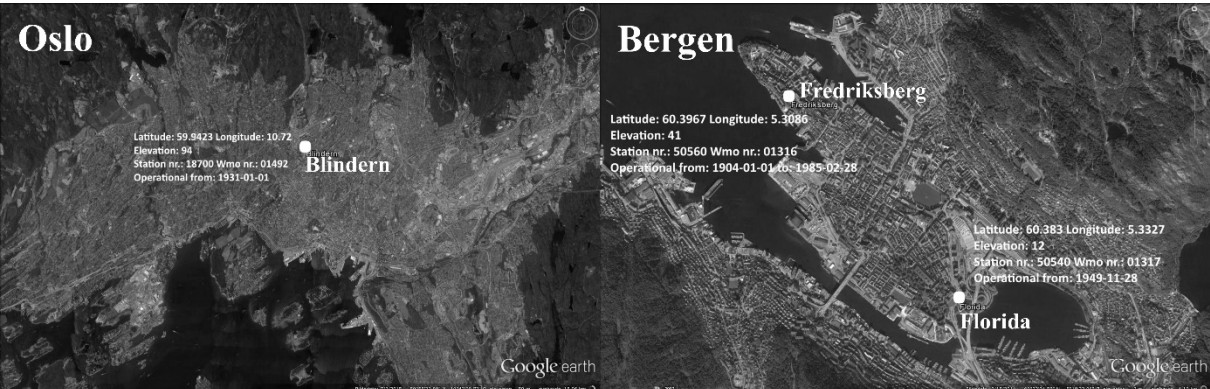

**Figure 1.** The locations of meteorological stations in Oslo and Bergen.

The yearly and monthly values for the climate wood rot index were calculated using the formulation from Reference [1]:

$$SI = \frac{\sum_{January}^{December} (T_{mean} - 2)(D - 3)}{16.7} \tag{1}$$

where $T_{mean}$ is the monthly mean temperature (°C) and $D$ is the mean number of days in the month with more than 0.25 mm (1 inch) precipitation ("precipitation days"). Reference [1] discussed in detail the conditions for use of the index, and Reference [2] suggested possible modifications for use in Norway but performed the index calculation with the similar formula as Reference [1] for the sake of inter-comparability, as also in this work. The climate index, temperature, and precipitation days for a period of years were calculated as the average of the annual values for the period.

## 3. Results

Figure 2 shows the calculated annual values for the climate wood rot index and the ranges of high, medium, and low climate risk as defined by Reference [1] and also used by Reference [2] and the annual averages for the temperature and the monthly number of days with more than 0.25 mm precipitation for the period 1961–2016.

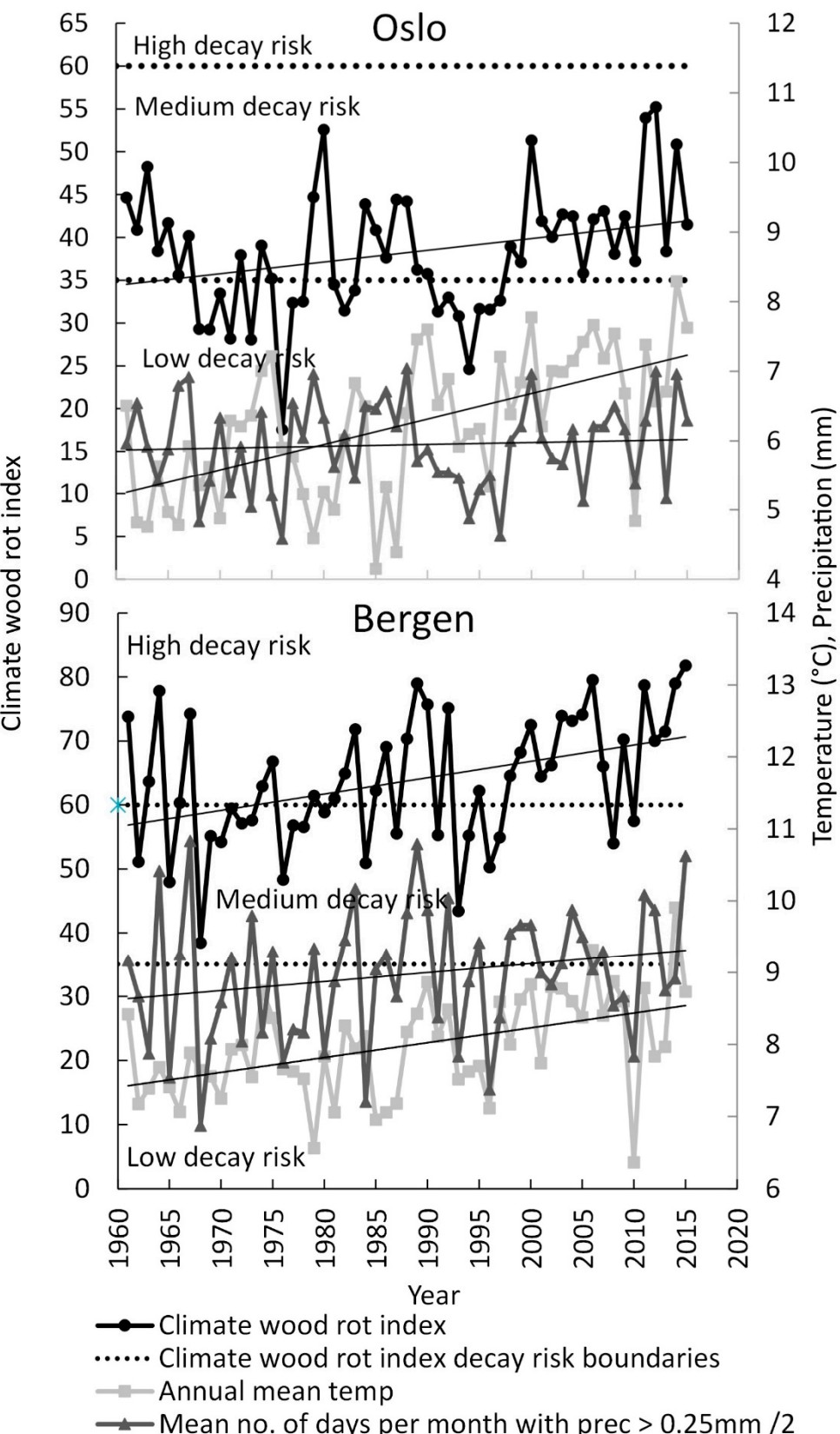

**Figure 2.** The change in the annual values for the climate wood rot index, average monthly mean temperature, and average number of days per month with precipitation over 0.25 mm in Oslo and Bergen from 1961–2016.

Clear positive trends were observed for all the three variables through the period, with higher values for the index and the number of precipitation days in the beginning, the approximate middle, and the end of the period. Table 1 shows the annual and seasonal values for the index calculated for 1961 and 2015 and the average values over the 55-year period from 1961–2015 and 30-year periods 1961–1990 and 1986–2015 calculated from the linear trends through the periods. The table also shows the absolute and relative changes in the index between 1961 and 2015 (55 years) and between the start (1961–1990) and end (1986–2015) periods (30 years).

**Table 1.** The annual and seasonal average values and changes for the climate index over the period 1961–2016.

| Oslo | Annual | Spring (M,A,M) | Summer (J,J,A) | Autumn (S,O,N) | Winter (D,J,F) |
|---|---|---|---|---|---|
| Climate index, 1961 | 34 | 4 | 22 | 9 | 0 |
| Climate index, 2015 | 42 | 7 | 27 | 8 | 0 |
| Climate index, 1961–2015 | 38 | 5 | 24 | 8 | 0 |
| Climate index, 1961–1990 | 37 | 5 | 23 | 9 | 0 |
| Climate index, 1986–2015 | 40 | 6 | 25 | 8 | 0 |
| Index change, 1961 to 2015 | 7 | 3 | 5 | 0 | 0 |
| Index change, 1961–1990 to 1986–2015 | 3 | 1 | 2 | −1 | 0 |
| % change, 1961 to 2015 | 21 | 65 | 25 | −3 | 0 |
| % change, 1961–1990 to 1986–2015 | 7 | 18 | 9 | −6 | 0 |
| **Bergen** | | | | | |
| Climate index, 1961 | 57 | 9 | 25 | 20 | 2 |
| Climate index, 2015 | 71 | 13 | 32 | 21 | 5 |
| Climate index, 1961–2015 | 64 | 11 | 29 | 21 | 4 |
| Climate index, 1961–1990 | 61 | 10 | 27 | 21 | 3 |
| Climate index, 1986–2015 | 67 | 12 | 30 | 20 | 4 |
| Index change, 1961 to 2015 | 14 | 4 | 7 | 1 | 2 |
| Index change, 1961–1990 to 1986–2015 | 6 | 2 | 3 | 0 | 1 |
| % change, 1961 to 2015 | 24 | 51 | 28 | 4 | 93 |
| % change, 1961–1990 to 1986–2015 | 9 | 21 | 11 | −2 | 30 |

The changes in the annual index over the 55 years were 8 for Oslo (increasing from a value of 34 to a value of 42) and 14 for Bergen (increasing from a value of 57 to a value of 71). This represents about a 20% change for both cities (21% in Oslo and 24% in Bergen) but an about 80% stronger linear trend (measured as the climate index change per year) from the higher index values in Bergen as compared to Oslo. The changes in the average periodic index values over the 30 years were 3 for Oslo (increasing from a value of 37 to a value of 40) and 6 for Bergen (increasing from a value of 61 to a value of 67), representing about 40% of the linear trend change between the start and end years.

The absolute changes in the index were largest in the summer. The relative changes were, however, larger in the winter (Bergen) and spring than in the summer. In the autumn, the index changes were small or negative (especially for Oslo) (Table 1).

Figure 3 shows the change in the average decadal climate wood rot index, temperature, and precipitation days from the decade before to the decade after any year.

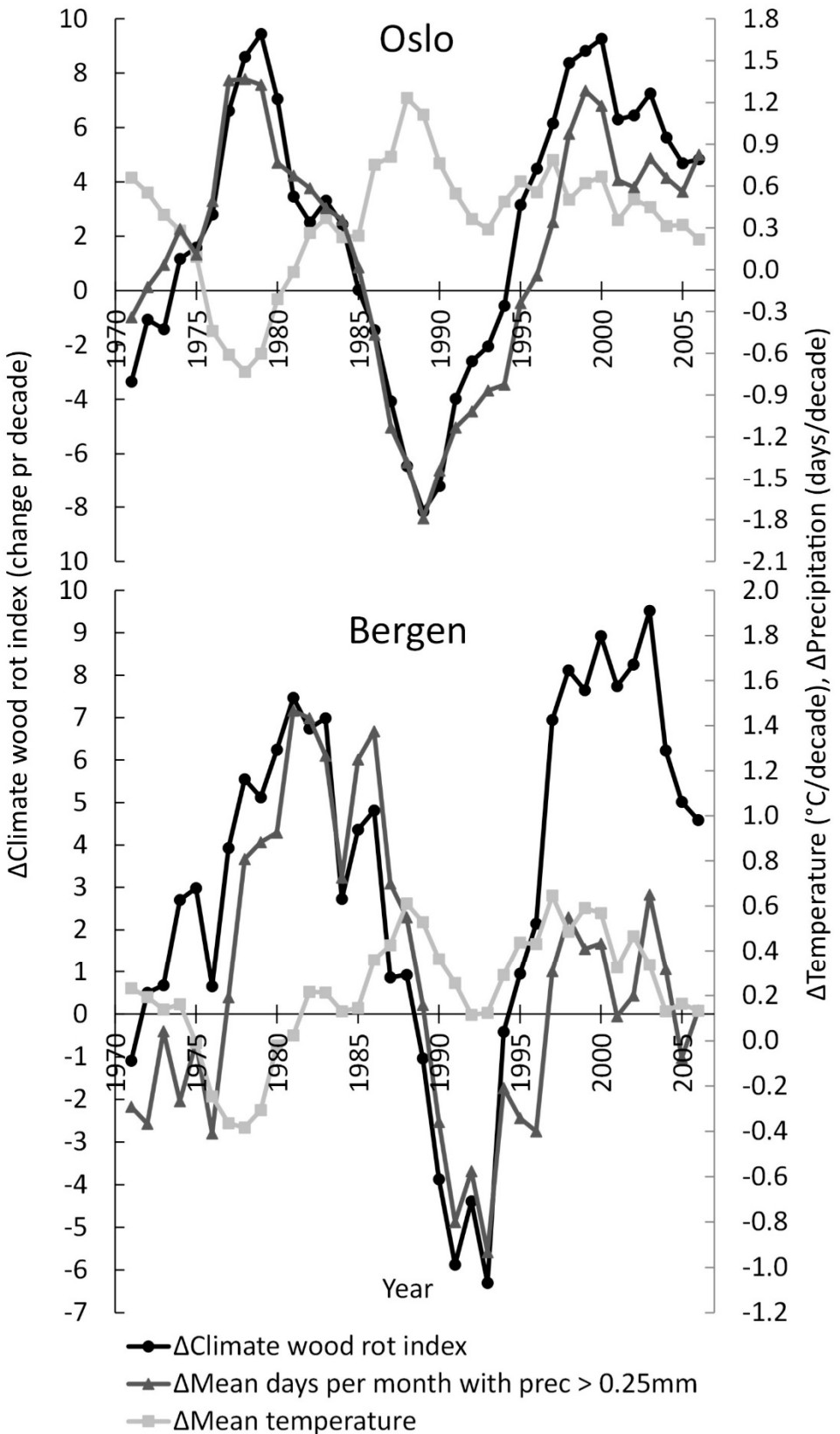

**Figure 3.** The change in the average decadal climate index, temperature, and precipitation days from the decade before to the decade after any year in Oslo and Bergen from 1961–2016.

The changes are positive in the start and end of the period from 1961 through to 2015 but negative in the middle of the period for both Oslo and Bergen. The change in the decadal index was closely accompanied by the change in the decadal precipitation days (ΔMean days per month with prec. > 0.25 mm), as could be expected from Equation (1). For Bergen, the change in the index is also mostly accompanied by the change in mean temperature, but this is not so for Oslo, as will be discussed below.

Figure 4 shows the average monthly values for the climate wood rot index, the temperature, and the precipitation days for the decades from 1961 ending with the 5.75 years period from 2011 until September 2016.

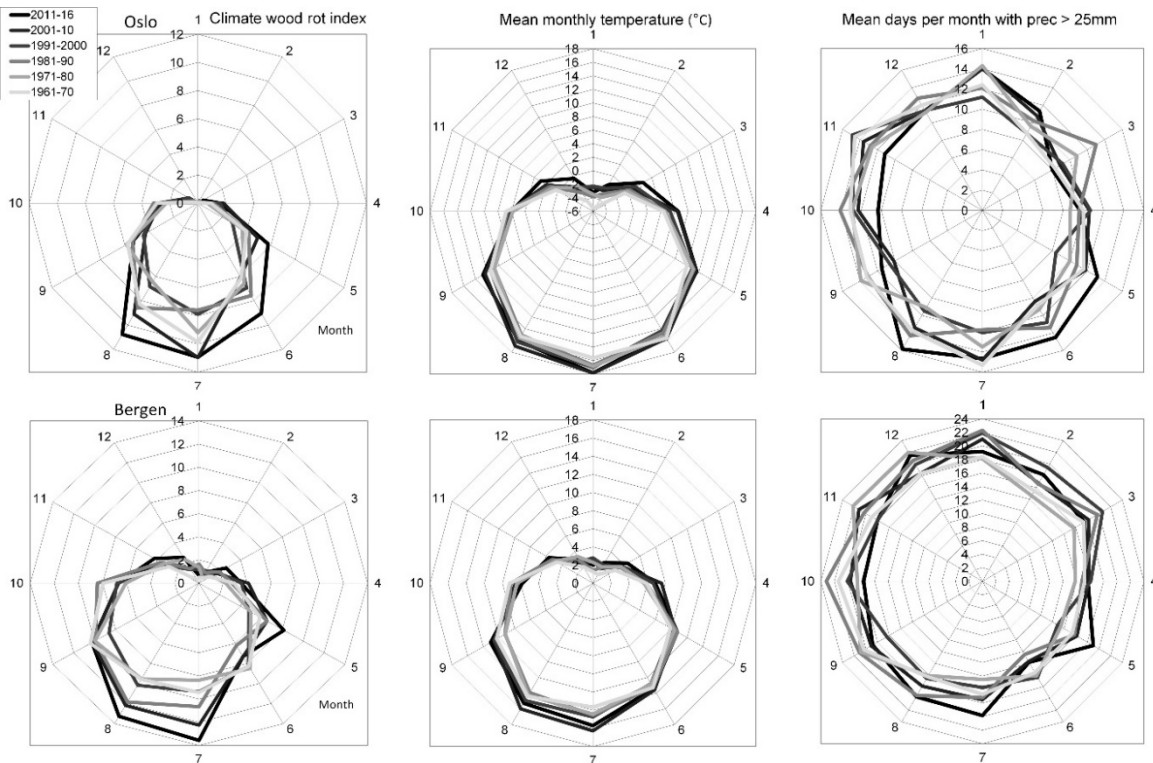

**Figure 4.** The change in the average values for the monthly climate index, mean temperature, and number of days with precipitation over 0.25 mm for the decades after 1961, ending with the period from 2011 through to September 2016, in Oslo and Bergen.

The climate index was generally increasing towards the later decades of the period from 1961 to September 2016 and mostly in the summer. Figure 5 shows the values for the linear trend coefficients for the monthly and yearly (see also Figure 2) climate wood rot index for the 1961–2016 and 1986–2016 periods. As an assistance to the interpretation of Figure 4, Figure 6 shows the linear trends for the monthly temperature and number of precipitation days from 1961 through to September 2016.

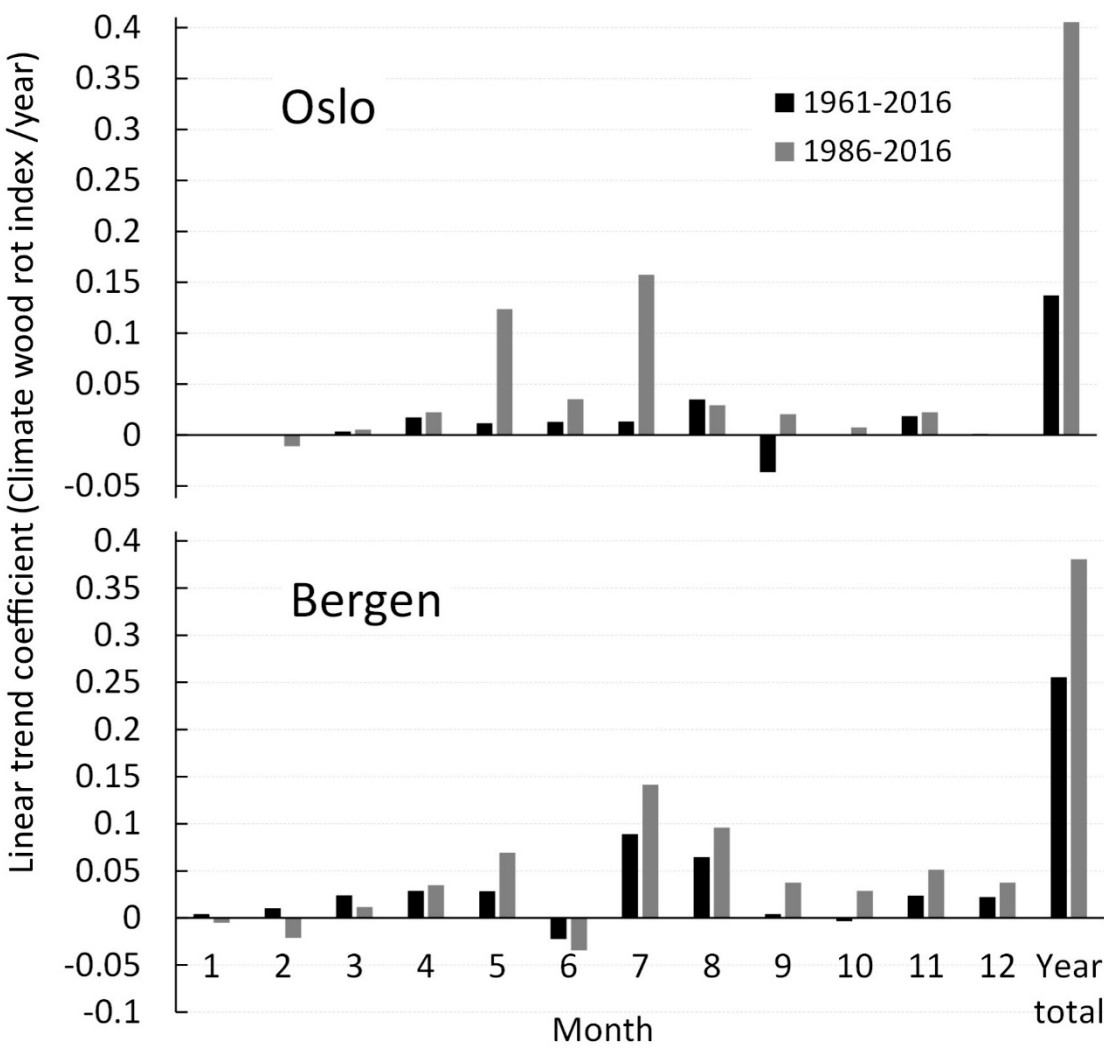

**Figure 5.** The linear trend coefficient values for the monthly and yearly climate index from 1961 and 1986 through to September 2016 in Oslo and Bergen.

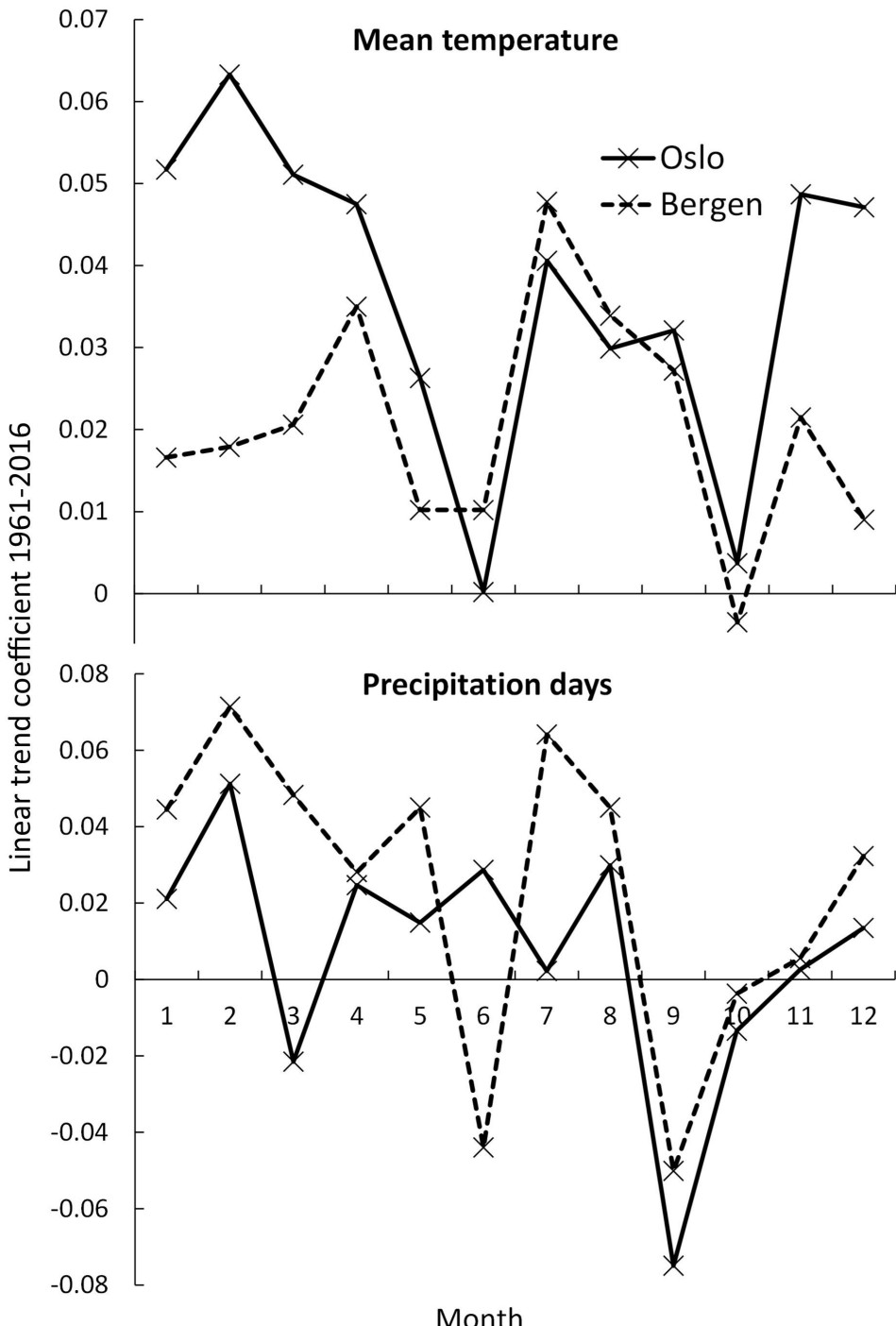

**Figure 6.** The linear trend coefficient values for the monthly mean temperature and the number of days per month with precipitation over 0.25 mm from 1961 through to September 2016 in Oslo and Bergen.

Figure 5 shows that the 1961–2016 linear trends for the months were mainly positive, as could be expected from the positive yearly trends, but with negative values for some months and periods. Higher values for the trends were generally observed in the later and shorter 1986–2016 period than the earlier and longer 1961–2016 period. The 1961–2016 trend in Oslo was strongest in August, clearly positive from April to July and in November, and low or zero in the other autumn and winter months. The trends in Bergen were strongest in July and August, generally about two times the values in Oslo, clearly positive from March to May and in November and December, and low or zero in the other autumn and winter months. Negative trends for September in Oslo and June and October in

Bergen corresponded with low values for the linear trends for the precipitation days in Oslo and the precipitation and temperature in Bergen (Figure 6).

## 4. Discussion

### 4.1. Climate Wood Rot Indices

Other rot indexes have been developed, which consider not only the wetting time but also the consequent moisture content in the wood needed for the fungi to grow [16]. The calculation of this index [16] is, however, considerably more complicated. This rot index takes into account the actual moisture content in the wood as dependent on the precipitation and wetting patterns and the moisture penetration. A modelling of the moisture infiltration in the wood, based on information about the wood characteristics and species, is needed. The integration of the moisture content over the penetration distance into the wood and a given time period (quarter year) is then performed. The result is interpreted as the "time of risk for fungal growth". Such more precise and detailed indexes are, however, difficult to apply to available climate data without more extensive investigations. Due to the simplicity of the "Scheffer's index", it is much easier to use and compare with other studies. Therefore, this study also uses Scheffer's climate index for the potential wood rot.

### 4.2. Climate Wood Rot Index Trends and Comparison with Climate Modelling

The evaluation of change in the wood rot climate index was made for locations of meteorological measurement stations in Oslo and Bergen with the longest available historical series of data. There is variation in the weather on long and short geographical scales. The use of data from other station/locations would probably have given somewhat different results for the trends. The applied meteorological data and estimated trends from the stations in Oslo and Bergen should be quite representative for the central built areas of these two largest cities in Norway. It could be hypothesized that similar trends, supporting the predictions from climate modelling, would be found from the evaluation of a larger set of historical measurement data from more stations. This work does, however, only report the evaluations based on measurement data from the central built areas of Oslo and Bergen.

The climate index for the decay potential of wood structures above ground was found to change in the period from 1961–1990 to 1986–2015 from a low/medium to medium decay risk in Oslo and from a medium/high to high decay risk in Bergen (Figure 2). Reference [2] reported the likely change in the index values for Oslo and Bergen, calculated from predicted values for temperature and precipitation obtained from climate change modelling. During the 60 years from the 1961–1990 to the 2021–2050 period, they predicted the change to become 9 (19%) for Oslo and 14 (20%) for Bergen. The observed change in the first 25 of these 60 years (Table 1), from the period 1961–1990 to the period 1986–2015, representing 42% of the time, was 33% (3/9) in Oslo and 43% (6/14) in Bergen of the total change predicted by Reference [2], which is close to the predicted trend (close to 42%). Reference [2] calculated somewhat higher absolute index values for Oslo and Bergen for the 1961–1990 period than in this work. The climate modelling for Norway shows similar expected trends over the country, with some variation [2,4,10].

The changes in the index indicate a larger increase in wood rot decay risk in the summer, then in spring and partly in November and December in both Oslo and Bergen (Table 1). The decadal changes (from past to the future decades) in the index were found to be closely followed by the decadal change in the number of precipitation days but not by the decadal changes in the temperature in the generally colder years in Oslo until 1993 and in Bergen until 1978. The reason for this may be the increase in temperatures below the level of 2 °C in the colder months in these years, which do not influence the index. Figure 6 clearly shows stronger positive temperature trends in the colder months, from 1961 through to September 2016, for Oslo than for Bergen. Still, the index for Oslo did not indicate wood rot risk in the winter in any year (see Table 1). Figure 4 shows the increasing temperature and that

a temperature of 2 °C was passed more often in autumn, spring, and winter in Bergen but not Oslo, when approaching the present.

This illustrates how the impact of climate change on wood decay depends on a threshold in the decay mechanism. The risks related to such possible transgressions of climate thresholds in relation to different impacts have been much debated. For wood rot, a certain amount of change at low temperatures have little influence, but when the temperature increases over a certain level, the monthly average of 2 °C, the rot becomes active and the decay impact increases. Thus, the expected effect of climate change on the rate and amount of wood rot in locations, which have historically had cold winters with freezing temperatures is to extend the duration of the rot progress to the spring and autumn months in relatively cold winter locations (e.g., Oslo) and even to the winter months in moderately cold locations (e.g., Bergen). The "winters" with no rot progress are expected to become shorter or to disappear depending on the level of the historical winter temperatures.

### 4.3. Climate Wood Rot Index Changes and Consequences

The maintenance of wooden buildings has become more demanding in a Norwegian climate, which has become warmer and wetter over the last 55 years, as indicated by the increase in the value for the wood rot index. A continuing similar future trend of increased rot risk can be predicted by applying the climate data from climate change modelling [4,17]. In this situation with increasing climate loads, it is important to provide better guidance and to include in standards, the improved procedures for climate adaptation during building construction and use. To limit the growth of fungi in wood, the most important measure is to reduce the wetting of and/or moisture in the wood. Indoors, the relative humidity should be kept below approximately 65% to avoid the growth of fungi [18].

With prolonged periods with rain in the future, it will be even more critical to protect construction and materials against surface wetting. In Norway, this will especially be the situation in the more rain-exposed coastal areas. Measures such as the reduction of the times construction is open to the weather during building, the increase in the dimensioning of gutters and the water runoff systems, and the application of improved coatings and surface treatments and/or their more frequent renewal are essential. In the less rain-exposed inland and eastern areas of Norway, the application of building principles previously more used in the wetter coastal areas may become more relevant. Such preventive modifications are expected to increase building and maintenance costs, which can, however, be minimized by adopting the most appropriate precautionary practices. Present experiences with adjustments in building and maintenance practice, due to the increased climate impact, are described in the literature [7,8]. Such implementation of adaptive practices is a huge ongoing effort.

## 5. Conclusions

The climate observations, recalculated in this work to the historical trend for the climate wood rot index, support the previous reported modeling predictions of continued changes in the direction of increasing rot risk for wooden buildings in Norway.

A change in the wood rot climate index since 1961 of about 20% was found for both Oslo and Bergen. The trend in the index (climate index change per year) was about 80% stronger in Bergen. The absolute index changes were largest in the summer and then in spring (50 to 60% of the summer increase) and small, zero or, even negative (autumn in Oslo) in the remaining seasons. The relative changes were higher in the spring than summer and very high in Bergen in the winter from a low value. The change to positive index values in the spring and winter happened as the monthly average temperatures increased to over 2 °C when the rot becomes active. This indicates extended periods through the year with temperature and humidity conditions favoring the growth of wood rot, an extended rot duration, and rot decay impact.

This study should raise awareness about increasing wood decay rates and related damage risks. Its conclusion further supports the increased focus on adaptation to reduce the consequences of the predicted climate change. More detailed and case-specific information about microclimates, materials,

and constructions are needed to decide on the concrete measures. Measures to reduce the humidity, moisture, and wetting of surfaces would be essential to prevent or reduce rot initiation and progress.

This study does not go further in giving such specific and practical advice. Such climate change adaptation studies for a "sustainable future" are, however, in great demand in, for example, European and Norwegian research programs. It would be important to obtain better dose-response (damage) understandings for the diverse impacts of climate change on buildings, including bio-growth and biological degradation. The existing and better-developed dose-response understandings would allow the estimation and mapping of historical changes and the comparison and prediction with climate model data on a local or wider geographical scale. An important field of study is the combination of such studies with local evaluations for single structures and buildings. The following questions are suggested to be important: Which essential recorded damages are described by available damage functions? How much variation in doses and damage responses are there due to microclimatic variations around structures, as compared to the larger geographical scale variations. How do the mechanisms combine to produce damages that are less well-described by existing damage functions? The general understanding of the direction of changes should be important to engage interest and to motivate preventive action. To decide which practical measures to apply, different, more case-specific, detailed, and concrete information is needed.

This study shows that the climate change that has happened over the past 55 years is according to the reported climate modelling and have increased the rot decay risk for wood structures in the cities of Oslo and Bergen in Norway. By itself, it gives no information about the possible future changes in such risks. This is a topic for ongoing climate modelling and for comparisons with the historical evidence, as was the topic for this study.

**Funding:** This research received no external funding.

**Acknowledgments:** The author wants to thank NILU-Norwegian Institute for Air research for giving budget time for this work.

**Conflicts of Interest:** The author declares no conflict of interest.

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
