# Peer review of "Observed Recent Change in Climate and Potential for Decay of Norwegian Wood Structures"

_climate, doi:10.3390/cli7020033_

Round 1

Reviewer 1 Report

 This study is about wood rot decay of structures and buildings in Norway. The topic is very interesting. I have some detailed comments listed below.

1, Line 12, Line 15, does this ‘climate index’ mean ‘wood rot climate index’ or is it about the general climate index? Need to make this clear and keep the consistency. In the main text, the authors could address ‘climate index’ indicates ‘wood rot climate index’ in this article.

2, Line 14, ‘the trend in the index was about 80% stronger in Bergen’, it is not clear here. What is the definition of ‘the trend’ here?

3, Line 17, 18, this statement should be rephrased or even removed. The previous sentence already addressed the index in the winter time is very low and it is higher in spring, so the relative changes were higher from winter to spring is needless to say. Or if you want to keep this, may add some numbers to specify the increase from which level to which level.

4, Line 19, how long is the ‘extended rot duration’? And what does the negative value mean?

5, Overall, the authors need to make the abstract more informative, and highlight the findings of this study.

6, The current discussion session is actually a part of the results. The conclusions session should discuss the limitation of this study, and the application of this study as well.

Author Response

Improvements were made according to the reviewer comments as follows:

1, Line 12, Line 15: ‘climate index’ was changed to ‘wood rot climate index’ in the two instances.

2, Line 14: The trend was now explained as “the trend (climate index change per year)”. This explanation was also given in line 135

3, Line 17, 18: The previous sentence only mention the absolute changes in semi-quantitative terms (larger/smaller). As absolute values for the index are not given in the abstract, the relative change are not determined directly by the absolute changes. Although it is reasonable to think that relative changes are larger in the winter and then spring this is not given: For example: a smaller change from a smaller value in spring could well give a similar relative change as in the summer, and likewise for the winter. The relative changes depend on the absolute index start values, as said by the reviewer, and the absolute changes. The absolute values could have been given, as suggested by the reviewer, to substantiate the reporting for the relative values. The author thinks the message would be the same as reporting the relative values directly:  Lower start values for the index in winter than spring, than summer, means larger relative changes. However, it was the intention to report in the abstract the changes (which is the basic result related to “climate change”), not the absolute values, which are reported later in the results and discussion – and which in a short abstract by themselves say less. Therefore the reporting of the relative changes are kept – as they are not () directly given by the absolute changes, and may be important for example in the winter when little change can mean rot initiation (see Discussion chapter)

4, Line 19: The sentence was changed to: “Change to positive index values in the spring and winter indicates temperature and humidity conditions favoring growth of wood rot and thus extended rot duration through the year”

In line 73 (previous 63) “decay potential” was changed with “wood rot decay potential”

5, Overall, abstract: The abstract need to be short (only about 200 words). The purpose of this paper was to evaluate if a previously reported expected change for this climate index (from climate modelling) could now be substantiated by actual measurement data. The conclusion about this is given in the abstract. This was the main intended message of the work, which is clearly communicated in the abstract, as supported by the evidences for the changes estimated/found for the wood rot climate index.

The abstract could be made more informative in other relevant respects, but not without increase in length. It was the wish to keep the focus of the abstract on the main conclusion. It seems to be little scope for extensions (according to editing guidelines: about 200 words). A sentence was added to the end of the abstract about consequences/adaptation:

“The expected increase in the future wood rot decay potential in Norway shows the need for increased focus on adaption measures to reduce related damages and costs.”

6. The Discussion was extended in several respects (see below “Request about Conclusion”). The author does not agree that the “present” (= previous) Discussion is part of “Results”. Chap 4.1 is closer to being “results”, but goes by including interpretation beyond pure results reporting. The argument being, paragraph by paragraph:

Chap 4.1, Paragraph 1 compared and discussed the similarity in the results from the previously reported (and referenced) climate index modelling and those from measured values uses in this work, showing similar results.

Chap 4.1, Paragraph 2 discussed why decadal changes in the index were closely followed by decadal change in the number of precipitation days, but not by the decadal changes in the temperature - in Oslo until 1993 and in Bergen until 1978. Discussing then that this may be due to more frequent changes in the monthly average temperature to above the index threshold value of 2°C after these years.

Chap 4.2, Paragraph 1 discussed how the maintenance of wooden buildings have become more demanding over the recent 55 years as indicated in this paper by the climate data, as earlier predicted by (referenced) climate modelling, and that climate modelling predicts further increase in the climate loads on buildings. (The purpose of this paper was not to discuss the validity of the climate modelling (which is a large topic) except comparing it with measured values), therefore no further discussion of the “future climate modelling” predictions are given). General preventive conservation and adaption measures that can limit the increase in humidity/moisture exposure and related rot decay are then suggested (discussed). The paper does not go beyond this to discuss particular measures, as this is a very large topic, which would in itself need a definition and limitation of cases. This is beyond the intent of this paper, and is good basis for much separate possible following assessments and work.

Chap 4.2, Paragraph 2 discussed, then, increased protection (adaptation) measures relevant for Norway, giving some general most important examples. againa: The intention of this paper was to compare previous modelling of the index with estimations base on measured climate values, not to discuss in more detail possibilities for adaptation. The reviewers’ interest in this is well understood, but it should be in different paper than this one.

Request about Conclusion:  

The author agrees that the Conclusion was not complete. The content was mainly a discussion of the consequences of reaching certain temperature thresholds in cold months.

This content (temperature thresholds in cold months) was found to fit better at the end of a Discussion chapter 4.2 Climate wood rot index trends and comparison with climate modelling, and was moved to there, keeping a short concluding text about this in the Conclusion.

The last three sentences from the Discussion were moved to the start of the Conclusion, as they were clearly better fitting/intended as concluding remarks.

The author agrees that the Conclusion was to restricted/short – in relation to the aspects mentioned by the reviewer (limitations and application) and also in relation to summarizing/concluding information, as also given in the abstract. This concluding information, was added to the Conclusion.

Paragraphs (no 3 and 4) about limitations of the study and applications and were added to the Conclusion.

Reviewer 2 Report

You have done a good work. I think some minors aspects should be modified in order to be published.

Describe briefly further work in conclusion 

How your work could be extended in other sectors /regions 

Author Response

The reviewer does not inform exactly about in what respect he/she finds that the method needs to be better described (?).

The Conclusion was revised to meet the requests of reviewer no.1. Some additional descriptions of possibilities for further research and how it could, potentially be extended in other sectors /regions, were added  to meet the requests of reviewer no.2. The author agrees this is an improvement giving some more perspective to the work. However, the author does not want to go into length about such topics, as they are not the main topics of this study, and they deserve much more attention by themselves.

Reviewer 3 Report

The manuscript should be improved in the scientific meaning, in addition to the results in each section.

e.g. The introduction section is poor of references and has no mention on rotting agents, method section have to be very clear (without discussion on used methods), the description of model urban areas should be broadly explained, the discussion section does not contain the comparison with other studies and root agents presentation.

Author Response

To initially give this a better explanation of the “the scientific meaning” of the work, to the start of the very first paragraph of the Introduction was added:

“The purpose of this work was to compare evidence available from meteorological measurements performed over the recent past, with reported modelling predictions for changes in wood rot decay risk in Norway.”

Answer to the directed comments:

“The introduction section is poor of references and has no mention on rotting agents”

Answer: The Introduction section has 12 references, related to the works evaluated through this new application of reported climate measurement data, and directly relevant for this. Following the advice of the reviewer, some further discussion of “rotting agents”, generally and related to the evaluation of Scheffer (ref. 1), was now included/cited in the Introduction:  (new ref 12 and 13) in lines 67-79.

“Method section have to be very clear (without discussion on used methods)”

Answer: The Discussion of methods (last paragraph in Materials and methods) was moved to the Discussion chapter. The methods paragraph was moved to before (swapped with) the paragraph with the data description.

The description of model urban areas should be broadly explained”

Measurement data, from single central meteorological stations in Oslo and Bergen were used in the calculations, no modelling data/results. Thus, the results are in principle only valid for the measurement locations. Outside of that, separate evaluations of representativeness would have to be made.

This was now made clearer by adding a first sentence to the data paragraph (no. 2) in the Materials and methods chapter:

Temperature and precipitation data from one meteorological station in Oslo and two stations in Bergen were used in the calculations.

I addition, a first paragraph in the Discussion sub-chapter “4.2: Climate wood rot index trends and comparison with climate modelling”, was now added explaining (discussing) this at some length. Even if there is no modelling area related to the work it was now explained (suggested)that:

 “The applied meteorological data and estimated trends from the stations in Oslo and Bergen should be quite representative for the central built areas of these two largest cities in Norway.”

 “The discussion section does not contain the comparison with other studies and rot agents presentation”

Some more information about rot agents was included in the introduction (see above). The main purpose of this study was comparison with previously reported climate modelling results for the same cities (Oslo and Bergen) now applying the actual observations (measurement data). The author doesn’t know about other published studies reporting this wood rot climate index for Oslo and Bergen (?). In the Introduction there is a summary [10] of the latest reporting of the historical and future expected Norwegian climate from the Norwegian Met. Inst.

Based on this it was added now in the Discussion that:

Climate modelling for Norway show similar expected trends over the country, with some variation [2,4,10].

Round 2

Reviewer 1 Report

This version is fine. I recommend the publication of this manuscript. 

Author Response

No further amendments are requested or were made by author as response to this review.

Reviewer 3 Report

The first paragraph from the Material and methods section of the first version has to be given back.

Author Response

The order of the two paragraps in the Materials and Methods chapter was changed. The first paragraph (data description) was swaped back to before the climate index model description. Like it was in the original submission.